# Oligomalt, a New Slowly Digestible Carbohydrate, Is Well Tolerated in Healthy Young Men and Women at Intakes Up to 180 Gram per Day: A Randomized, Double-Blind, Crossover Trial

**DOI:** 10.3390/nu15122752

**Published:** 2023-06-15

**Authors:** Odd Erik Johansen, Delphine Curti, Maximilian von Eynatten, Andreas Rytz, Anirban Lahiry, Frederik Delodder, Gerhard Ufheil, Carmine D’Urzo, Audrey Orengo, Kate Thorne, Jaclyn S. Lerea-Antes

**Affiliations:** 1Nestlé Health Science, 1000 Lausanne, Switzerland; maximilian.eynatten@nestle.com (M.v.E.); carmine.durzo@nestle.com (C.D.); kate.thorne@nestle.com (K.T.); 2Société des Produits Nestlé, 1000 Lausanne, Switzerland; delphine.curti@rdls.nestle.com (D.C.); audrey.orengo@rdls.nestle.com (A.O.); 3Nestlé Research, Clinical Research Unit, 1000 Lausanne, Switzerland; andreas.rytz@rdls.nestle.com (A.R.); anirban.lahiry@rd.nestle.com (A.L.); frederik.delodder@rdls.nestle.com (F.D.); 4Nestlé Research and Development Konolfingen, Société des Produits Nestlé S.A., 3510 Konolfingen, Switzerland; gerhard.ufheil1@rd.nestle.com; 5Nestlé Product Technology Center NHS, Société des Produits Nestlé S.A., Bridgewater, NJ 08807, USA; jaclyn.lerea-antes@us.nestle.com; 6Nestlé Health Science, Bridgewater, NJ 08807, USA

**Keywords:** clinical trial, dietary carbohydrate, food, gastrointestinal tolerance

## Abstract

In this randomized, double-blind triple-crossover study (NCT05142137), the digestive tolerance and safety of a novel, slowly digestible carbohydrate (SDC), oligomalt, an α-1,3/α-1,6-glucan α-glucose-based polymer, was assessed in healthy adults over three separate 7-day periods, comparing a high dose of oligomalt (180 g/day) or a moderate dose of oligomalt (80 g/day in combination with 100 g maltodextrin/day) with maltodextrin (180 g/day), provided as four daily servings in 300 mL of water with a meal. Each period was followed by a one-week washout. A total of 24 subjects (15 females, age 34 years, BMI 22.2 kg/m^2^, fasting blood glucose 4.9 mmol/L) were recruited, of whom 22 completed the course. The effects on the primary endpoint (the Gastrointestinal Symptom Rating Score (GSRS)) showed a statistically significant dose dependency, albeit of limited clinical relevance, between a high dose of oligomalt and maltodextrin (mean (95% CI) 2.29 [2.04, 2.54] vs. 1.59 [1.34, 1.83], respectively; difference: [−1.01, −0.4], *p* < 0.0001), driven by the GSRS-subdomains “Indigestion” and “Abdominal pain”. The GSRS difference ameliorated with product exposure, and the GSRS in those who received high-dose oligomalt as their third intervention period was similar to pre-intervention (mean ± standard deviation: 1.6 ± 0.4 and 1.4 ± 0.3, respectively). Oligomalt did not have a clinically meaningful impact on the Bristol Stool Scale, and it did not cause serious adverse events. These results support the use of oligomalt across various doses as an SDC in healthy, normal weight, young adults.

## 1. Introduction

Carbohydrates represent a large family of heterogenous molecules derived from carbon, hydrogen, and oxygen that can be naturally occurring or derived. Dietary carbohydrates, also called “glycemic carbohydrate”, are a major macronutrient for both humans and omnivorous animals, and it is recommended that adults in western countries obtain approximately half their daily caloric intake from dietary carbohydrates. This is illustrated by a variety of dietary guidelines, such as The Dietary Guidelines for Americans, which recommend that carbohydrates make up 45–65% of total daily calories in people aged two years old and older [1]. In other regions of the world, in particular in Asia, the percentage consumed in reality can be even higher [2]. While the absolute minimum dietary requirement for glycemic carbohydrates depends on the amount of fat and protein also being ingested, the European Food Safety Authority (EFSA) recommends an intake of 130 g/day to cover the need of glucose for the brain, with a minimum of 50 to100 g per day to prevent ketosis [3]. Of the ingested carbohydrates, approx. 60% is in the form of polysaccharides, mainly starch, whereas the disaccharides sucrose and lactose contribute approximately 30% and 10%, respectively [4]. Monosaccharides (such as glucose and fructose) are naturally present in fruits and found in manufactured foods and drinks.

Digestible carbohydrates are an important source of energy for most mammalian cells, including as the primary source of fuel for the brain. They are also used as sweeteners (mostly sucrose, glucose, and fructose) and to add texture, bulk, and increase appearance and preservation/shelf life of many foods [4]. However, excess ingestion of sugars and carbohydrates that are rapidly digested can lead to a number of unfavorable metabolic alterations, e.g., increasing triglycerides, free fatty acids, and lipids [5,6,7,8], and are associated with several health conditions (e.g., gout, fatty liver, obesity, type 2 diabetes mellitus, cardiovascular disease). Therefore, several scientific interest groups in the field of nutrition have issued recommendations on the amount of added sugar to be consumed; these include the World Health Organization (WHO), which recommends an upper limit of free sugars at 10% of calories, with an ultimate goal of reducing sugars consumption to 5% of calories [9], and the American Heart Association Nutrition Committee, which recommends that women and men consume no more than 100 and 150 kcal of added sugar per day, respectively [10].

A growing interest in the health and food industry is therefore focused on the use of complex dietary carbohydrates that are slowly digested and can substitute, or at least reduce the amount of, the less-desired rapidly digestible carbohydrates. Unlike fiber, defined by the WHO and Codex Alimentarius as all carbohydrates that are neither digested nor absorbed in the small intestine and which have a degree of polymerization (DP) of ten or more monomeric units [11], slowly digestible carbohydrates (SDC) are completely digested, and they are defined as “carbohydrates that are likely to be completely digested in the small intestine but at a slower rate” [12]. Since SDC, due to their structural complexity, are fully but slowly digested throughout the small intestine, their net digestion profile should result in a less-pronounced glycemic response, coupled with a lower insulin requirement [13].

There have been several innovations and developments that have aimed to identify an appropriate SDC [14], which may have multiple applications for use, e.g., to replace more rapid carbohydrates in food matrices while still being thermally stable and compatible with liquid nutrition drinks and enteral formula manufacturing. Novel α-glucans, a large group of linear, branched, or cyclic oligo- and polysaccharides of D-glucose, where the glucose moieties are joined by either [α-D-Glc*p*-(1↔2)-α-D-Glc*p*], [α-D-Glc*p*-(1↔3)-α-D-Glc*p*], [α-D-Glc*p*-(1↔4)-α-D-Glc*p*], or [α-D-Glc*p*-(1↔6)-α-D-Glc*p*] glycosidic linkages [15], represent one such class of SDC.

This study evaluated the tolerability of oligomalt, a novel SDC (formerly also known as alternan saccharide (AS 15)) with a DP of approximately 15 relative to a rapidly available carbohydrates (maltodextrin), in healthy individuals. Since it is anticipated that oligomalt is slower to digest than a readily available carbohydrate (such as corn syrup or maltodextrin), albeit still fully digestible [16], this study aimed to specifically investigate the GI tolerability of oligomalt when taken sub-chronically.

## 2. Materials and Methods

### 2.1. Study Design and Participants

In this single-centre, randomized-controlled, double-masked, three-period, three-intervention, cross-over study, participants consumed two different doses of the investigational product, or a control product, for 7 consecutive days. The study protocol was approved by the Institutional Review Board of Orange County Research Center, USA, and Temasek Polytechnic, Glycemic Index Research Unit, Singapore, and the study was carried out in compliance with the Harmonized Tripartite Guideline for Good Clinical Practice from the International Conference on Harmonisation [17] and the Declaration of Helsinki [18].

### 2.2. Inclusion and Exclusion Criteria

Eligible volunteers of any gender were enrolled in the study if, following informed signed consent, they fulfilled the inclusion criteria. The key inclusion criteria were as follows: male or female, self-reported to be healthy, age 18–65 years, BMI 18.5–29.9 kg/m^2^, and having access to a smartphone with android or iOS version compatible with “Patient Cloud application”. Key exclusion criteria included fasting plasma glucose ≥6.1 mmol/L at screening, type 1 or type 2 diabetes, known food allergy or intolerance to the test product, any treatment with medications known to affect gastric motility, or any condition known to affect gastro-intestinal integrity and food absorption.

A further exclusion criterion was the habitual consumption of more than four servings per day of high-fiber food or an extreme dietary habit low in carbohydrates. The former was pragmatically assessed with the habitual dietary fiber intake short food frequency questionnaire (DFI-FFQ) [19], which derives a score (in grams) based on the participant’s recall of average servings over the last year based on ingestion of fruit, vegetables, breads and cereals, nuts and seeds, and legumes (scores > 35 g/day was considered high), while the latter was based on three elimination questions (1: Do you eat any serving of any fruit during a typical week? 2: Do you eat/drink any serving containing sugar during a typical week (soft drinks, candy, juice, sports drink, chocolate, cakes, buns, pastries, ice cream and breakfast cereals)? 3: Do you eat any serving of any starch during a typical week (bread, pasta, rice, potatoes, French fries, potato chips, porridge, muesli, beer, vegetables etc.)?) and if the answer was yes to any one of these, they were not considered to be undertaking extreme dieting.

Moreover, participants who had any score of a “severe” symptom for any of the symptoms included in the Gastrointestinal Symptom Rating Scale (GSRS), a validated questionnaire exploring the presence and severity of gastrointestinal (GI) symptoms over the previous 7 days [20,21], were excluded. A full list of in-/exclusion criteria is provided in the Appendix A.

### 2.3. Study Conduct and Schemes

During the different 7-day sequences with investigational product consumption (Schematic study illustration in the Appendix A), participants consumed either 180 g oligomalt/day (as 45 g 4 times daily), 80 g oligomalt (as 20 g 4 times daily) plus 100 g maltodextrin (as 25 g 4 times daily), or 180 g maltodextrin alone (as 45 g 4 times daily). The total daily dosages to be tested reflected a high daily dose (i.e., 180 g/day), e.g., the quantity an individual may receive from a sole source of nutrition (enteral nutrition) or a realistic serving dose (i.e., 45 g) for a meal-replacement product or oral nutritional supplement. To understand the GI effect of oligomalt specifically, the overall carbohydrate amount was kept constant while the amount of oligomalt and easily digested maltodextrin were adjusted to reflect the desired oligomalt dose provided. The products were consumed under free living conditions outside of the research facility.

Given that the caloric load was 180 kcal per serving (or 720 kcal total/day), all participants received general nutritional recommendations and were asked to pay specific attention to their consumption of grains and cereals and sugar-rich foods and beverages since the study contributed to an excess intake of energy.

### 2.4. Interventional Products

The maltodextrin used in this study was “Glucidex^®^ 40” (Roquette Frères, Lestrem, France), a maltodextrin with a dextrose equivalent equal to 40. Oligomalt, the novel α-glucan polymer [α-D-Glc*p*-(1↔3)-α-D-Glc*p*], [α-D-Glc*p*-(1↔6)-α-D-Glc*p*], was produced in collaboration with Evoxx technologies GmBH (Monheim am Rhein, Germany) as previously described [16]. In short, oligomalt is produced via transglycosylation reaction catalyzed by the enzyme, alternansucrase, with sucrose as a donor of one glucose moiety and maltose as the acceptor substrate. This reaction results in a linear oligosaccharide with alternating α1-6 and α1-3 linkages.

The oligomalt used in this study was comprised of small quantities of free sugars (<5.5% sugars, of which <0.2% is fructose and 5.3% is leucrose but zero glucose, sucrose, isomaltose, or maltose), with an average ± standard deviation DP of 16.5 ± 0.1 and a molecular weight of 2.7 ± 0.02 kDa.

The test products were provided as sachets (i.e., powders with approx. 4% moisture), each containing 45 g of carbohydrates (either 45 g oligomalt, 20 g oligomalt + 25 g maltodextrin, or 45 g maltodextrin), which were to be dissolved in 300 mL water and taken together with a meal.

### 2.5. Endpoints

The predefined primary objective of this trial was to investigate the gastrointestinal tolerability of oligomalt when consumed daily for 7 consecutive days at two different dosages (80 g/day in combination with 100 g of maltodextrin/day and 180 g/day, respectively) as compared to maltodextrin (180 g/day), based on the GSRS questionnaire [20,21]. The effects on GSRS subdomains and the relationship between the intervention period, or intervention sequence, and GSRS were analysed post hoc. The secondary objectives of this trial were to investigate gastrointestinal tolerability based on daily stool evaluation and assessed for each bowel movement and stool frequency, by the Bristol Stool Scale questionnaire [22,23]. In addition to GI tolerability, any adverse events were captured over the course of the study.

### 2.6. GSRS

The GSRS is a 15-item instrument designed to assess the symptoms associated with common GI disorders. It has 5 subscales (reflux, diarrhea, constipation, abdominal pain, and indigestion). Responses to the items range from 1 to 7, with higher scores representing more discomfort (1: No discomfort at all; 2: Slight discomfort; 3: Mild discomfort; 4: Moderate discomfort; 5: Moderately severe discomfort; 6: Severe discomfort; 7: Very severe discomfort). It has been used in multiple studies across multiple conditions and is well validated [20,21]. The GSRS delivers one overall score and five different subdomain scores. The latter is obtained by combining different questionnaire items into a reflux score (average score of items 2 and 3), diarrhea score (average score of items 11, 12, and 14), constipation score (average score of items 10, 13, and 15), abdominal pain score (average score of items 1, 4, and 5), and indigestion score (average score of items 6, 7, 8, and 9). The total GSRS score is the average of all five sub-scores.

### 2.7. Bristol Stool Scale

The Bristol Stool Scale is a scale that classifies stools, ranging from the hardest to the softest, and defines 7 types of stool: hard (types 1–2), normal (types 3–5), and loose stools (types 6–7) [22,23]. The Bristol Stool Scale also indirectly reflects gut transit time [24], and the instrument has been widely used in different populations and been found to have high validity and reliability [25].

### 2.8. Sample Size Estimation

The sample size was determined based on previous safety studies carried out with other SDCs on feasibility considerations at the investigational site, as well as considerations pertaining to not exposing participants unnecessarily to study products and study procedures. Thus, no formal sample size calculation was performed; however, a total of *n* = 20 participants completing the study was considered necessary to provide meaningful insights for longer-term tolerability and GI digestibility of oligomalt. Accounting for an expected dropout of a maximum of 15%, we aimed to recruit 24 participants.

### 2.9. Blinding and Randomisation

Each participant consumed all three test products in a sequence (Appendix A) according to a crossover design. To account for potential systematic bias, all 6 possible sequences were used. Randomization was performed using iMedidata RTSM. The research staff and the participants were unaware of the assigned intervention sequence.

### 2.10. Product Administration

The investigational product and the comparator were packed as individual doses of 45 g, and each individual was to take the product 4 times daily for one week. Participants were asked to dissolve the powder in 300 mL of water at room temperature and were also asked to take at least 3 of their 4 daily doses with a meal at timepoints 08:00 ± 2 h, 12:00 ± 2 h, 16:00 ± 2 h, and 20:00 ± 2 h (minimum time interval between individual product intakes was ≥3 h). Upon ingestion of the last dose on day 7, a minimum washout of 7 days was required before the next intervention period began, in random sequence, as determined by the randomization system (3 × 3 crossover).

### 2.11. Data Collection

The participant-reported outcomes. i.e., questionnaires, were collected using the iMedidata patient cloud (i.e., in a remote set up), and each participant received a unique activation code that needed to be used to generate their personal unique pin. Participants were asked to report the following on a daily basis: product compliance, GSRS, and Bristol Stool Scale.

### 2.12. Statistical Analysis

The effects of the interventions on the GSRS (overall effect, subdomain effects, and effects related to intervention period) and Bristol Stool Scale were assessed using a linear mixed model, and they accounted for information from all 7 days of intervention. A *p*-value of <0.05 was conventionally considered significant. Analysis of baseline characteristics, adverse events and stool frequency according to Bristol Stool Scale, as well as the post hoc analysis of GSRS by product and intervention sequence, were summarized descriptively.

## 3. Results

The first participant was screened 16 August 2021, and the third intervention sequence ended on 11 October 2021. A total of 24 subjects were enrolled, randomized, and had at least one investigational product intake. Two subjects dropped out, which resulted in 22 subjects completing all periods and sequences, reflecting a high adherence to study procedures and product compliance. The reasons for participants dropping out were that one person had an adverse event after the first period (inflammation of the knee), which was not related to the product, and the other withdrew from the study without further explanation. All 24 participants were included in the analysis set both for efficacy and safety.

The population, with mean age 34 years, was healthy (Table 1) by intention, reflected by BMI 22.2 kg/m^2^, systolic/diastolic blood pressure 119/78 mmHg, and fasting blood glucose 4.9 mmol/L at screening. None of the included participants habitually ate high volumes of fiber or followed any extreme diet.

### 3.1. Effects on GSRS

Figure 1 depicts the overall response to the GSRS, i.e., regardless of the intervention period (i.e., week 1, week 2, or week 3) or intervention sequence (i.e., which of the products that was introduced first, second, and third), and Figure 2 depicts the GSRS subdomain scores. As can be seen, a slight oligomalt dose dependency for the GSRS was observed, with a statistically significant difference between maltodextrin and the highest oligomalt dose (oligomalt 180 g/day mean (95% CI) 2.29 [2.04, 2.54] vs. maltodextrin 180 g/day 1.59 [1.34, 1.83], difference: [−1.01, −0.4], *p* < 0.0001) but not between the intermediate oligomalt dose (oligomalt 80 g + maltodextrin 100 g/day 1.84 [1.58, 2.09] and maltodextrin 180 g/day 1.59 [1.34, 1.83], difference: [−0.55, 0.05], *p* = 0.1024). The clinical significance of this result is considered to be limited.

The subdomain scores of the GSRS (Figure 2) showed that reflux (oligomalt 180 g/day 1.50 [1.24, 1.77], oligomalt 80 g + maltodextrin 100 g/day 1.26 [0.99, 1.52], maltodextrin 180 g/day 1.32 [1.06, 1.57]), constipation (oligomalt 180 g/day 1.84 [1.52, 2.17], oligomalt 80 g + maltodextrin 100 g/day 1.41 [1.09, 1.73], maltodextrin 180 g/day 1.35 [1.03, 1.66]), and abdominal pain (oligomalt 180 g/day 1.82 [1.51, 2.14], oligomalt 80 g + maltodextrin 100 g/day 1.63 [1.31, 1.95], maltodextrin 180 g/day 1.47 [1.16, 1.79]) were within the reference range (~1.4,. 1.6, and 1.4, respectively [26]). There were no significant differences for reflux between any dose of oligomalt and maltodextrin (*p* > 0.05); whereas a statistically significant but not clinically relevant differences between the highest oligomalt dose and maltodextrin for constipation (difference [−0.89, 0.10], *p* = 0.0156) and abdominal pain (difference: [−0.67, −0.04], *p* = 0.0289) was seen. There was also a significantly higher score for the highest dose of oligomalt group vs. maltodextrin alone for diarrhea (oligomalt 180 g/day 2.38 [1.97, 2.80], oligomalt 80 g + maltodextrin 100 g/day 1.97 [1.56, 2.38], maltodextrin 180 g/day 1.46 [1.06, 1.86]; difference: [−1.45, −0.4], *p* = 0.0009). With the normative score for diarrhea ~1.4 [26], this indicates a tendency for more loose stools with the high dose of oligomalt. Similarly, a significantly higher score for indigestion with the highest dose of oligomalt vs. maltodextrin alone was observed (oligomalt 180 g/day 3.30 [2.84, 3.76], oligomalt 80 g + maltodextrin 100 g/day 2.49 [2.03, 2.95], maltodextrin 180 g/day 2.08 [1.63, 2.53]; difference: [−1.76, −0.67], *p* < 0.0001), but all groups appeared to score higher than normative data (~1.8) for the indigestion subdomain.

The corresponding overall GSRS value (regardless of intervention) by period (i.e., week 1, week 2, or week 3) is depicted in Figure 3, and it indicates a dynamic pattern with a higher overall GSRS score for the first intervention period (2.08 [1.84, 2.33]) than the second (1.98 [1.73, 2.23]) and third periods (1.65 [1.40, 1.90]), indicating a physiologic GI adaption to all products with time. There was a significant difference between period 1 and 3 (difference: [0.13, 0.74], *p* = 0.0062) and between period 2 and 3 (difference: [0.02, 0.63], *p* = 0.0370). Of note, within period 3, the overall GSRS score was within the reference range (~1.6–1.7) [26].

This observation was also supported by more granular analysis assessing intake of oligomalt or maltodextrin by intervention sequence (Figure 4); for examples, in people who received oligomalt 180 g/day as their third intervention sequence (i.e., after having received maltodextrin 180 g/day and oligomalt 80 g + maltodextrin 100 g/day in the preceding weeks), the mean ± standard deviation GSRS was 1.6 ± 0.4, which was numerically no different from their baseline GSRS (1.4 ± 0.3).

### 3.2. Effects on Stool Consistency

Figure 5 displays the effects on stool consistency, and as indicated, there were no clinically relevant differences in Bristol Stool Scale across intervention groups, which were all considered to be within the normal range of 3–5 [23] (oligomalt 180 g/day 4.43 [4.13, 4.74], oligomalt 80 g + maltodextrin 100 g/day 4.47 [4.16, 4.77], maltodextrin 180 g/day 4.08 [3.79, 4.38]). However, statistically significant differences were noted between both the 80 g oligomalt + maltodextrin 100 g/day group and the maltodextrin 180 g/day group (difference: [−0.7, −0.07], *p* = 0.02), as well as the 180 g/day oligomalt and maltodextrin 180 g/day group (difference [−0.66, −0.04], *p* = 0.03), but without clinical relevance.

Stool frequency by product and intervention sequence (Figure 6), obtained from the number of entries into the Bristol Stool Scale questionnaire, also supports this observation.

### 3.3. Safety and Adverse Events

Adverse events (Table 2) occurred in 60.9% of the subjects while receiving the highest 180 g/day oligomalt dose, 56.5% during the 80 g/day dose, and 29.2% in the maltodextrin group alone, with higher proportions dose-dependently considered to be related to oligomalt. The numerical higher number of adverse events with oligomalt was driven by the system organ class “Gastrointestinal disorders”, where a numerical imbalance was seen for the preferred terms “Flatulence”, “Abdominal distension”, and “Abdominal pain”, which align with the results of the GSRS. There were no reports of serious adverse events or adverse events leading to discontinuation.

## 4. Discussion

This study, in 24 healthy adult volunteers (15 females, age 34 years, BMI 22.2 kg/m^2^, fasting blood glucose 4.9 mmol/L) with a moderate to low pre-enrollment fiber intake, investigated the 7-day tolerability and safety of ingestion of a novel SDC, oligomalt, ingested at high or moderately high doses. We observed that oligomalt ingested four times daily, either at 180 g/day or 80 g/day, was reasonably well tolerated relative to the ingestion of 180 g/day of maltodextrin; it did not cause serious adverse events and demonstrated high compliance. Of note, the study population had a GI adaptation to oligomalt, which was reflected by an overall higher 7-day GSRS in the highest dose group and driven by subdomain scores for indigestion and abdominal pain, which were ameliorated over time, and by the intervention period, reaching norm values during in last period [25].

In consideration of a “minimal important difference” (MID), which is the smallest difference in the scores that is perceived as significant by the clinician or the patient [26], this has not uniformly been defined for GSRS, although the MID for GSRS in renal transplant recipients has been proposed around 0.6 for abdominal pain, 0.8 for reflux, 0.4 for diarrhea, 0.7 for indigestion, and 0.7 for constipation [25]. As such, taking into account the adaptation that occurs, there does not seem to be a clinically significant difference between high dose of oligomalt and maltodextrin. The higher proportion of reported adverse events, which was driven by GI adverse events, particularly with the highest dose, probably also reflects this phenomenon, i.e., the overall numbers of adverse events reflect a higher number of events with early onset and do not adequately account for the observed period where GSRS is ameliorated. The clinical relevance of the early increase in GSRS is likely not clinically significant and is in line with expectations in a population with a low pre-intervention intake of fiber products when exposed to a SDC [27,28], where an adaptation is expected.

The evolutive the GSRS pattern, with tolerability developed over time, therefore underscores the product’s tolerability and safety profile, and is also supported by a previous investigation of oligomalt that found that a 33 g dose was completely metabolized as assessed with a hydrogen breath test over 4 h [16].

Another observation from this study was that there was no clinically meaningful differential effect on stool consistency according to the Bristol Stool Scale. However, the high-dose oligomalt resulted in a slight signal for increase in loose stool, aligned with higher GSRS subdomain diarrhea score. Similar to the tolerability, the Bristol Stool Scale score was subject to a GI adaptation, as indicated by the intervention sequence analysis that found that the stool frequency was numerically lower at the third sequence compared to the first.

### Limitations and Strengths

The key limitation of this study was that exposure to individual products did not extend beyond 7 consecutive days. However, in reality, participants were exposed to oligomalt for at least 14 days of the overall 21 days of intervention, i.e., over two study sequences, albeit with different doses and a washout period, which provides reassurance with respect to longer-term tolerability. Additionally, this study was a single-center study, involving a limited number of participants of predominantly Caucasian origin who did not have any underlying health conditions, including inflammatory bowel disease or gastrointestinal issues, at relatively young age. Blood or other biomarkers which could be relevant, e.g., gut microbiota, were also not collected nor analyzed. Furthermore, oligomalt was consumed with water only; whereas in reality, oligomalt will likely be consumed in a complex food matrix, which usually translates to better tolerability [29] (although the participants were instructed to consume the test drinks with meals during the study). Given that the product, when taken with water in this study, was reasonably well tolerated, particularly after some GI adaption, it is conceivable that it would be at least as well tolerated in a food matrix. The main strength of this study is its use both a high and intermediate dose of oligomalt to ensure tolerability at a range of intake doses. The high-dose oligomalt was particularly high, resulting in favorable tolerability, and supports a high tolerability window.

## 5. Conclusions

This study evaluated the GI tolerability of a novel SDC, oligomalt. Oligomalt was well tolerated when taken at a moderate (80 g/day) or high (180 g/day) dose, with no effect on stool frequency. An evolutive pattern with decreasing GI symptoms was seen with exposure over time, with normative scores being reached following 2–3 weeks of exposure to oligomalt. Given oligomalt’s healthier composition, it appears to be a potential viable alternative carbohydrate source for nutritional products, when also taking into account the implications for blood glucose, the insulin response, and overall metabolic health.

## Figures and Tables

**Figure 1 nutrients-15-02752-f001:**
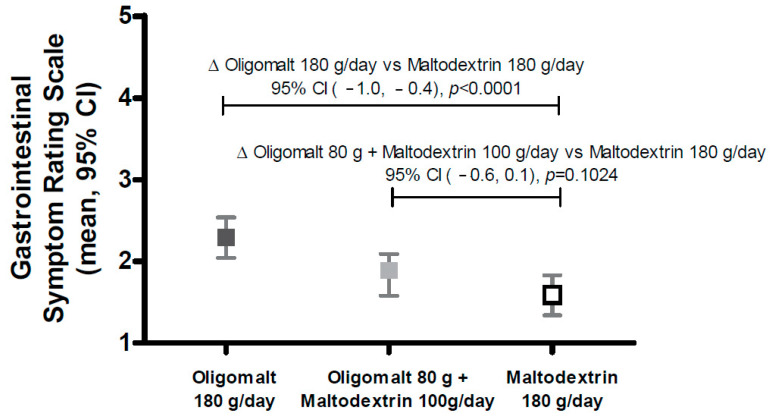
Effect on total score for Gastrointestinal Symptom Rating Scale by intervention, regardless of intervention sequence. The figure shows the mean (95% CI) (*Y*-axis), and the differences between different interventions are expressed with 95% CI and *p*-values. Abbreviations: g—day; CI—confidence interval.

**Figure 2 nutrients-15-02752-f002:**
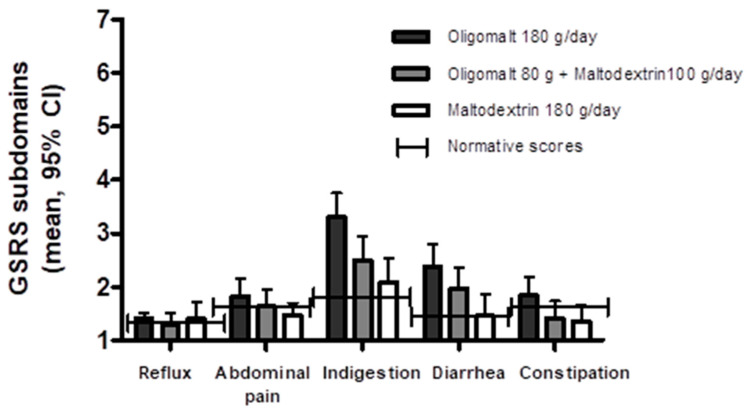
Subdomain scores for the Gastrointestinal Symptom Rating Scale (GSRS) by intervention, regardless of intervention sequence. The figure shows the mean (95% CI) (*Y*-axis) as well as the published GSRS subdomain normative scores. Abbreviations: g—day; CI—confidence interval.

**Figure 3 nutrients-15-02752-f003:**
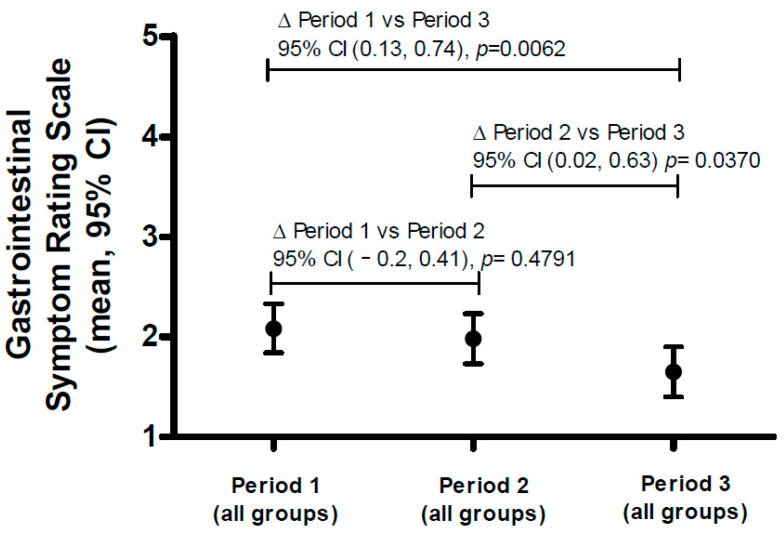
Total score for Gastrointestinal Symptom Rating Scale by period regardless of intervention. The figure shows the mean (95% CI) (*Y*-axis), and the differences between different periods are expressed with 95% CI and *p*-values. Abbreviations: CI—confidence interval.

**Figure 4 nutrients-15-02752-f004:**
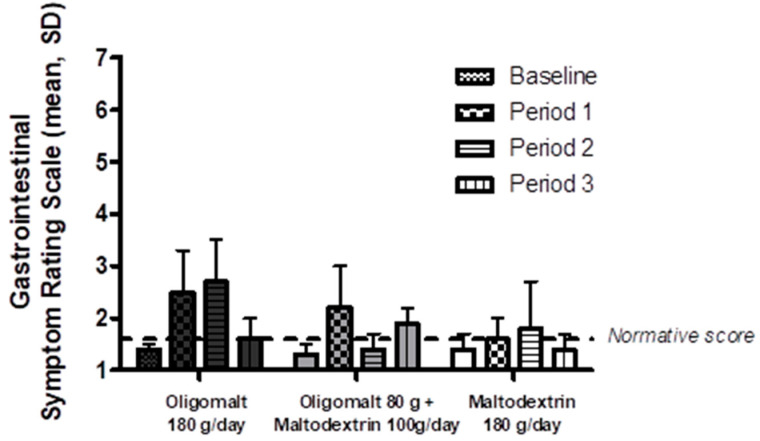
Total Gastrointestinal Symptom Rating Scale by intervention and intervention period. The figure shows the mean Gastrointestinal Symptom Rating Scale (SD) (*Y*-axis) over the three treatment periods by intervention and published normative score. Baseline denotes score at V1. Abbreviations: g—day; SD—standard deviation.

**Figure 5 nutrients-15-02752-f005:**
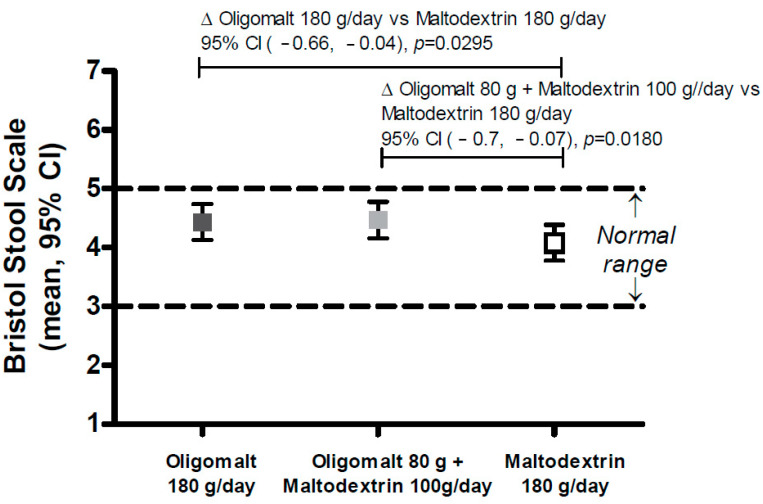
Bristol Stool Scale by intervention in the overall population (FAS). The figure shows the mean (95% CI) (*Y*-axis) as well as the published Bristol Stool Scale normative scores. Abbreviations: g—day; CI—confidence interval.

**Figure 6 nutrients-15-02752-f006:**
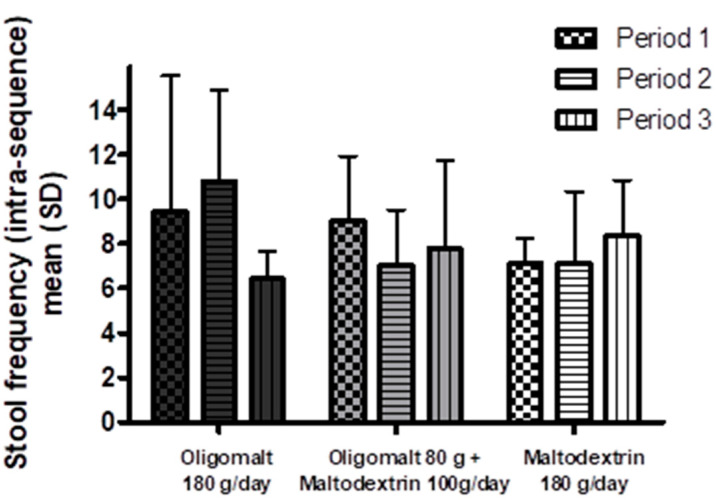
Stool frequency by intervention and period. The figure shows mean stool frequency (SD) (*Y*-axis) over the three treatment periods by intervention. Abbreviations: g—day; SD—standard deviation.

**Table 1 nutrients-15-02752-t001:** Baseline characteristics of the study population: *n* (%) or mean (SD).

Description	*n* (%), or Mean (SD)
N	24 (100%)
Age, years	33.7 (9.02)
Sex, female/male	15 (62.5%)/9 (37.5%)
Weight, kg	65.3 (11.85)
Body Mass Index, kg/m^2^	22.2 (2.64)
Heart rate, beats per minute	69.6 (12.28)
Systolic Blood Pressure, mmHg	118.5 (14.54)
Diastolic Blood Pressure, mmHg	78.4 (10.07)
Blood glucose, mmol/L	4.9 (0.44)

Abbreviations: SD—standard deviation; kg—kilogram; mmHg—millimeter of mercury.

**Table 2 nutrients-15-02752-t002:** Adverse events (AEs). Safety analysis set: subjects with events (%).

	Oligomalt 80 g + Maltodextrin 100 g	Oligomalt 180 g/Day	Maltodextrin 180 g/Day
N	23	23	24
Subjects with ≥1 AE	13 (56.5%)	14 (60.9%)	7 (29.2%)
Subjects with ≥1 serious AE	0 (0%)	0 (0%)	0 (0%)
Subjects with AE by worst severity			
Mild Moderate Severe	3 (13%) 7 (30.4%) 3 (13%)	3 (13%) 8 (34.8%) 3 (13%)	3 (12.5%) 4 (16.7%) 0 (0%)
Subjects with ≥1 product related AE			
Probable Unlikely Unrelated	8 (34.8%) 2 (8.7%) 3 (13%)	12 (52.2%) 0 (0%) 2 (8.7%)	4 (16.7%) 0 (0%) 3 (12.5%)
AEs by System Organ Class (SOC) and Preferred Terms (PT)			
GI disorders	10 (43.5%)	12 (52.2%)	3 (12.5%)
- Abdominal distension	2 (20%)	9 (75%)	1 (33.3%)
- Abdominal pain	1 (10%)	4 (33.3%)	0
- Abdominal pain upper	0	0	1 (33.3%)
- Constipation	2 (20%)	1 (8.3%)	1 (33.3%)
- Diarrhea	2 (20%)	1 (8.3%)	
- Feces discolored	1 (10%)	0	0
- Flatulence	1 (10%)	5 (41.7%)	1 (33.3%)
- Gastric disorder	1 (10%)	0	0
- GI pain	1 (10%)	1 (8.3%)	0
- GI sounds abnormal	0	1 (8.3%)	0
- Nausea	1 (10%)	2 (16.7%)	1 (33.3%)
- Vomiting	1 (10%)	0	0
General disorders and administration site conditions	1 (4.3%)	0	0
Infections and infestations	3 (13%)	0	1 (4.2%)
Metabolism and nutrition disorders	1 (4.3%)	0	0
Musculoskeletal and connective tissue disorders	0	2 (8.7%)	0
Nervous system disorders	0	3 (13%)	2 (8.3%)
Reproductive system and breast disorders	1 (4.3%)	1 (4.3%)	0
Respiratory, thoracic, and mediastinal disorders	2 (8.7%)	0	1 (4.2%)
Skin and subcutaneous tissue disorders	0	0	1 (4.2%)

## Data Availability

Original data supporting these results are available on request to the corresponding author for reasonable purposes.

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
