# Peer review of "Oligomalt, a New Slowly Digestible Carbohydrate, Is Well Tolerated in Healthy Young Men and Women at Intakes Up to 180 Gram per Day: A Randomized, Double-Blind, Crossover Trial"

_nutrients, 2023, doi:10.3390/nu15122752_

Round 1
Reviewer 1 Report
Beautifully written and conducted study. My only comment is that the conclusion should say that there were no adverse effects noted in healthy, normal weight young adults.
I assume the effects for other ages and body types would be the same but there is no way of assuring that is true.
Should there be a note of caution about people with IBS or other GI issues around the use of this product?
Author Response
Comments and Suggestions for Authors
Beautifully written and conducted study.
Author’s response
We thank you for this positive overall comment and thank the Reviewer for taking time to review and provide feedback.
My only comment is that the conclusion should say that there were no adverse effects noted in healthy, normal weight young adults.
Author’s response
We thank you for this constructive comment. We have amended the conclusion to state:
“These results support the use of oligomalt across various doses as a SDC in healthy, normal weight, young adults.”
I assume the effects for other ages and body types would be the same but there is no way of assuring that is true.
Author’s response
We agree that we cannot extrapolate our results to individuals with other characteristics beyond those studied here, but as an information to the reviewer we are currently doing studies in people with type 2 diabetes as well as in people with overweight or obesity. As a response to this valid comment we have expanded the limitation section by specifying this:
“Additionally, this study was a single-centre study, involving a limited number of participants of predominantly Caucasian origin, who did not have any underlying health conditions, including inflammatory bowel disease or gastrointestinal issues, at relatively young age”
Should there be a note of caution about people with IBS or other GI issues around the use of this product?
Author’s response
We agree that this is a topic that need careful consideration. In other SDC products, where fructose levels are higher, this definitively is a concern. Studies in adults for example, comparing 3 doses of fructose (15, 25 and 50 g), found that 100% of healthy volunteers could absorb 15 g of fructose, 90% could absorb 25 g of fructose, but only 20–30% could absorb 50 g (Rao SS et al. Ability of the normal human intestine to absorb fructose: evaluation by breath testing. Clin Gastroenterol Hepatol. 2007;5:959–63). In pediatrics, appropriate dosage appear to be even lower where a maximal dose of 10–15 g has been suggested (Jones HF et al. Developmental changes and fructose absorption in children: effect on malabsorption testing and dietary management. Nutr Rev. 2013;71:300–9). Since oligomalt contains a very limited amount of fructose, which in this study in the high dose group cumulative is < 10g, any potential disturbances would likely not be related to this. Also, since oligomalt also is completely digested (Lamothe LM., et al., Effects of α-D-glucans with alternating 1,3/1,6 α-D-glucopyranosyl linkages on postprandial glycemic response in healthy subjects. Carbohydrate Polymer Technologies and Applications, 2022;4:100256), we believe that this is not likely. Nevertheless, we agree that we only would know this with the appropriate studies and have therefore added these considerations to the limitation section.
“Additionally, this study was a single-centre study, involving a limited number of participants of predominantly Caucasian origin, who did not have any underlying health conditions, including inflammatory bowel disease or gastrointestinal issues, at relatively young age”
Reviewer 2 Report
This is a well executed study on oligomalt and the potential benefits of slowly digestible carbohydrates (SDC) in the light of long term CVD risk prevention due to improved post-prandial insulin/glucose/TG responses, as well as GI tolerability. The manuscript is professionally written. There is a well written Supplementary document.
As is I have no further comments.
Author Response
Comments and Suggestions for Authors
This is a well executed study on oligomalt and the potential benefits of slowly digestible carbohydrates (SDC) in the light of long term CVD risk prevention due to improved post-prandial insulin/glucose/TG responses, as well as GI tolerability. The manuscript is professionally written. There is a well written Supplementary document.
As is I have no further comments.
Author’s response
We thank you for these positive comments about the study design and the Reviewer for taking time to review and provide feedback and comments.